# Utility of three-dimensional echocardiography for evaluating right ventricular size and function and ventricular myocardial deformation in repaired tetralogy of fallot

Kwannapas Saengsin[1]*, Kamonchanok Intamul[1], Pakpoom Wongyikul[2,3], Phichayut Phinyo[2,3], Rekwan Sittiwangkul[1], Seehapong Petcharat[4], Jinnawat Rattanang[5], Patanin Chindarungrueangkun[5], Ashwin Prakash[6]

1 Division of Cardiology, Department of Pediatrics, Faculty of Medicine, Chiang Mai University, Chiang Mai, Thailand, 2 Center for Clinical Epidemiology and Clinical Statistics, Faculty of Medicine, Chiang Mai University, Chiang Mai, Thailand, 3 Department of Biomedical Informatics and Clinical Epidemiology (BioCE), Faculty of Medicine, Chiang Mai University, Chiang Mai, Thailand, 4 Division of Cardiology, Department of Pediatrics, Faculty of Medicine, Khon Kaen University, Thailand, 5 The Northern Thailand Heart Center, Faculty of Medicine, Chiang Mai University, Chiang Mai, Thailand, 6 Department of Cardiology, Boston Children's Hospital, Harvard Medical School, United States of America

* Kwannapas_09@hotmail.com

## Abstract

### Purpose

Right ventricular (RV) dysfunction remains a major long-term complication in patients with repaired Tetralogy of Fallot (rTOF). While cardiac magnetic resonance (CMR) is the gold standard for right ventricular (RV) assessment, it is limited by accessibility and cost. Three-dimensional echocardiography (3DE), with its strain imaging capabilities, offers a promising alternative for the serial evaluation of right ventricular (RV) size and function. This study aims to compare RV volumetric and myocardial deformation parameters obtained by 2D and 3D echocardiography (2DE, 3DE) with RV function measured by CMR in patients with rTOF.

### Materials and Methods

We retrospectively analyzed 43 patients with rTOF who underwent same day 2D, 3D echocardiography, and CMR between November 2023 and December 2024. RV volumes, ejection fraction (RVEF), and global longitudinal strain (GLS) were measured across modalities. Correlation, Bland-Altman analysis, and area under receiver operating characteristic (AUC) curves were used to evaluate the agreement and diagnostic accuracy for detecting right ventricular (RV) systolic dysfunction.

**Data availability statement:** Data cannot be shared publicly because of institutional regulation. Data are available from the Institutional Research Ethics Committee of Faculty of Medicine, Chiang Mai University (contact via email: Researchmed@cmu.ac.th) for researchers who meet the criteria for access to confidential data.

**Funding:** The Faculty of Medicine Endowment Fund for Medical Research, Chiang Mai University, Thailand, provided support for this work. The funding source had no influence on the study's design, conduct, data collection, management, analysis, interpretation, manuscript preparation, review, approval, or the decision to submit for publication.

**Competing interests:** The authors have declared that no competing interests exist.

## Results

Three-DE demonstrated a strong correlation with CMR for RVEDV (r = 0.95) and RVEDVi (r = 0.91), with a mild underestimation. RVEF by 3DE was significantly lower than CMR (46.4 ± 8.8% vs. 50.3 ± 7.3%, p = 0.002). RV-GLS values differed across modalities, with 3DE yielding more negative values than 2DE (−20.3 ± 4.4% vs. −18.5 ± 4.7%, p = 0.009). Among patients with CMR-RVEF <48%, both 2D and 3D Echo-derived RV-GLS were significantly reduced.

## Conclusion

Three-dimensional echocardiography demonstrates a consistent association with CMR for the assessment of right ventricular volumes but modest underestimation of volumetric and functional parameters. Abnormal right ventricular strain was observed in patients with CMR-defined systolic dysfunction, supporting the clinical relevance of strain analysis. Overall, 3DE may serve as a feasible complementary tool for longitudinal right ventricular assessment in patients with rTOF, alongside CMR as the reference standard.

## Introduction

Tetralogy of Fallot (TOF) is one of the most common congenital heart diseases (CHD), with long-term sequelae affecting right ventricular (RV) and left ventricular (LV) function. While surgical correction enhances early outcomes, long-term risks such as right ventricular dysfunction, pulmonary regurgitation, and left ventricular deterioration persist, posing significant challenges to patient management [1]. Assessing ventricular function in these patients is key to improving prognosis and determining the appropriate timing for intervention [2,3].

Among imaging modalities, cardiac magnetic resonance imaging (CMR) is considered the gold standard for non-invasive evaluation of ventricular volumes, function, and myocardial strain due to its high spatial resolution and reproducibility [1,4,5]. However, its use is hindered by financial constraints and limited accessibility, especially in developing countries where such advanced imaging modalities are not readily available. As a result, standard echocardiography remains the primary tool for follow-up assessments, although it provides a less comprehensive analysis of right ventricular function [6,7].

Three-dimensional echocardiography (3DE) has gained recognition as a viable alternative due to its real-time imaging capability, portability, and lack of ionizing radiation [8,9]. The global and regional myocardial strain assessment, particularly using speckle-tracking echocardiography and evaluated ventricular size and function, provides valuable insights into subclinical myocardial dysfunction [10]

Although 2D and 3D echocardiography and CMR have been individually validated for evaluating ventricular performance, few studies have directly compared these imaging techniques in assessing right ventricular function and myocardial strain in

patients with TOF. This research aims to analyze the agreement between 3DE and CMR in evaluating RV volume and strain to establish the utility and dependability of echocardiography in long-term monitoring. Increasing accessibility to GLS technology may be essential to support its integration into clinical practice and enhance its validation through comparison with the gold-standard CMR.

The primary objective of this study is to evaluate the utility of three-dimensional echocardiography (3DE) in the assessment of right ventricular (RV) size, function, and global longitudinal strain (GLS) in patients with rTOF, using CMR as the reference standard.

Specifically, we aim to:

1. Compare RV volumetric measurements obtained from 3D echocardiography with those derived from CMR to assess agreement and measurement accuracy.

2. Assess and compare RV global longitudinal strain (GLS) measured by 2D and 3D echocardiography and evaluate their correlation with CMR-derived RV ejection fraction (RVEF) as a marker of systolic function.

3. Evaluate the diagnostic performance of strain parameters (from 2D and 3D echocardiography) in detecting RV systolic dysfunction, defined by CMR-RVEF < 48%.

## Materials and methods

### Study population

A cohort of all TOF patients was retrospectively enrolled from our institution's cardiology clinic records between 01/11/2023 and 31/12/2024. In accordance with the echocardiography protocol for TOF patients undergoing same-day CMR, the inclusion criteria encompassed both pediatric and adult patients who had undergone complete surgical repair of their TOF and were scheduled for CMR by the pediatric cardio outpatient clinic. Patients with poor quality 3DE imaging, significant residual intracardiac shunts, significant arrhythmias, or contraindications to CMR, such as implanted metallic devices, were excluded from the study.

### Ethical approval process and inform consent

This study was approved by the Institutional Review Board of the Faculty of Medicine, Chiang Mai University, on 24/06/2025 (Approval ID: PED-2568–0373, date of approval 24/06/2025, and date of expiration 23/06/2027). Data were accessed for research purposes on 30/06/2025. The study was conducted in accordance with the Declaration of Helsinki (2024), which sets ethical principles for research involving human subjects, including those related to anonymity and data protection. This study was conducted as a retrospective observational study using clinically acquired imaging data collected between November 2023 and December 2024. All echocardiography and cardiac magnetic resonance imaging examinations were performed as part of routine clinical care according to an institutional protocol. Ethical approval was obtained from the Institutional Review Board of the Faculty of Medicine, Chiang Mai University, to permit retrospective access, analysis, and publication of the data.

### Imaging protocols

**Echocardiographic assessment.** *Two-Dimensional Echocardiography (2DE):* All echocardiograms were performed using a Phillips Epiq 7 with 5 or 8 Mhz probes (Philips Medical Systems) according to patient size. The analysis was performed on the cardiac cycle with the highest image quality among three. The software delineated the RV endocardial contours from the basal septum through the apex and along the free wall to the tricuspid valve which were manually traced and adjusted to ensure complete myocardial coverage. TomTec Imaging Systems (Unterschleissheim, Germany) was used to divide the RV into six segments for automatic tracking of myocardial motion. RV GLS was computed as a

predefined weighted average of strain values from six regions: three located on the septum and three on the free wall, spanning basal, mid, and apical levels [11].

*Three-Dimensional Echocardiography (3DE)*: Three-dimensional speckle-tracking strain analysis was performed at an average frame rate exceeding 30 frames per second. A matrix-array 3D transducer was positioned in a slightly modified four-chamber view to ensure comprehensive visualization of the right ventricle. Adjustments in total gain and time gain compensation were made to optimize delineation of the endocardial border, with additional rotational and translational maneuvers used to visualize both inflow and outflow tracts [12]. Offline analysis of right ventricular volumes was conducted using TomTec software (TomTec Imaging Systems, Unterschleissheim, Germany). The software generated three orthogonal planes, identifying frames at end-diastole and end-systole, and automatically displayed four-chamber, short-axis, and coronal views. Manual tracing was done at the level of the papillary muscle bases. Right ventricular end-diastolic (3D-RVEDV) and end-systolic volumes (3D-RVESV) were measured, and the 3D right ventricular ejection fraction (3D-RVEF) was calculated using the formula: [(RVEDV – RVESV)/ RVEDV] × 100. Volume curves of the RV were also analyzed [13,14].

**Cardiac magnetic resonance imaging.** Cardiac magnetic resonance imaging was performed using a 1.5T MRI scanner (Siemens, Erlangen, Germany) with steady-state free precession (SSFP) cine sequences for volumetric assessment. The cine sequence was a breath-hold 2D B-TFE, using retrospective cardiac triggering and providing 30 heart phases per RR interval. Specific imaging parameters included: TR/TE = 2.8/1.42, bandwidth 1078 Hz, FOV 30 × 30 cm2, voxel size is 1.5x1.5mm with 8 mm slice thickness, interpolated to 0.75 × 0.75 × 8 mm3 through zero-filling. An imaging acceleration method (Compressed Sense) was used to accelerate the acquisition by a factor of 1.5. Breath-hold duration was 11 seconds, depending on cardiac triggering rate (in 2/3/4-chamber views, axial and short-axis stack, and sagittal oblique views across the RV outflow tract). The cardiac function was evaluated by a single experienced imaging pediatric cardiologist (KS).

## Blinding and image analysis

Two-dimensional (2D) and 3DE and CMR were performed on the same day to minimize physiologic variability. However, all post-processing analyses were conducted on separate days. Echocardiographic analyses, including 2D right ventricular strain and 3D right ventricular volumetric and strain measurements, were performed by an investigator who was blinded to the CMR results at the time of analysis. Likewise, CMR post-processing and interpretation were performed independently and without knowledge of the echocardiographic findings. This blinding strategy was implemented to minimize observer bias and ensure objective comparison between imaging modalities.

## Definition

- Right ventricular GLS was measured on obtained RV GLS by averaging RV free wall and septal values [15].

- Right ventricular systolic dysfunction, defined by an RVEF < 48% [16].

## Data analysis and statistical methods

All statistical analyses were conducted using Stata 17 (StataCorp, Collegue station, Texas, USA). Categorical variables were summarized as frequencies and percentages, while numerical data were assessed for distribution using histograms and described as means with standard deviations (SD) or medians with interquartile ranges (IQR), depending on their distribution. A paired t-test was used for continuous variables comparison as appropriate. A p-value less than 0.05 was considered statistically significant.

Pearson correlation was used to assess the relationship between cardiac parameters. The correlation coefficient ranges from −1 to +1 and was interpreted as follows: negligible (0.00–0.10), weak (0.10–0.39), moderate (0.40–0.69), strong (0.70–0.89), and very strong (0.90–1.00) [17]. Measurement error of 3D echo to CMR was assessed by calculating

the mean bias, limit of agreement (LOA), and P10 (defined as the percentage of echocardiogram values within ±10% of CMR values) [18]. Bland-Altman plot was illustrated to describe the pattern of measurement error [19]. Parameters from echocardiogram and CMR were also assessed for their ability to discriminate patients with RVEF<48% using area under receiver operating characteristic (AUC). An AUC of 0.70–0.80, 0.80–0.90, and above 0.90 were considered acceptable, excellent, and outstanding, respectively [20].

Sample size was calculated based on a pilot study of 30 patients. The anticipated absolute mean bias between RVEF measured by echocardiogram and CMR was 5, with a standard deviation of 8. Assuming an intra-subject correlation coefficient of 0.5, 29 patients were required to achieve 90% power at a 5% alpha level. Calculations were performed using a two-sample paired-means test.

## Results

### Baseline characteristics

A total of 46 patients with repaired Tetralogy of Fallot (rTOF) were initially enrolled in the study. However, three were excluded due to suboptimal 3DE image quality, which failed to capture the entire right ventricle owing to marked dilation (CMR-derived RVEDVi>200 mL/m²) (Fig 1). Among 43 patients, 27 (62.8%) were male, with a mean age of 17.9±5.8 years. Most patients had TOF with pulmonary stenosis (88.4%), while pulmonary atresia was observed in 9.3%. Palliative shunts were performed in 32.6% of patients, predominantly modified Blalock–Taussig shunts (78.5%). Definitive surgical repairs were completed in first operation of life in 67.4% of the cohort, mainly using transannular patches (79.1%). The median age at total correction was 40 months (IQR: 31, 49 months). Syndromic associations occurred in 9.3% of patients. No patient in our cohort had heart failure symptoms (Table 1).

### The agreement and correlation between 3DE and CMR in the Right Ventricular Size and Function

Three-dimensional echocardiography (3DE) demonstrated a strong correlation with CMR for right ventricular end-diastolic volume (RVEDV) (r=0.95, p<0.001) and right ventricular end-diastolic volume index (RVEDVi) (r=0.91, p<0.001), with a mean bias of −16.3 mL (LOA: −55.4, 22.9 mL) and −11.3 mL/m² (LOA: 40.8, 18.2 mL/m²), respectively. Right ventricular end-systolic volume (RVESV) and right ventricular end-systolic volume index (RVESVi) also showed strong correlation (r=0.90–0.88), though the bias was 0.8 mL (LOA: −33.2, 31.7 mL) and −2.0 mL/m² (LOA: −22.2, 18.3 mL/m²), respectively. The right ventricular ejection fraction (RVEF) measured by 3DE was significantly lower than CMR (46.4±8.8% vs. 50.3±7.3%, p=0.002) (Table 2, Fig 2, and Supplementary Table 1).

### The Comparison and Correlation between 2D and 3D Echocardiographic Right Ventricular Strain Parameters in Patients with rTOF

When comparing RV strain parameters between 2D and 3D echocardiographic techniques, the 3D RV global longitudinal strain (GLS) values were significantly more negative than those obtained by 2D echocardiography (–20.3±4.4% vs. –18.5±4.7%, P=0.009), with a moderate correlation (r=0.52). Similarly, RV septal longitudinal strain (RV-LS septal) measured by 3D echocardiography showed more negative values compared to 2D (–19.9±5.0% vs. –17.8±4.9%, P=0.008), with a correlation coefficient of 0.48. Although the 3D RV free wall strain (RV-LS free wall) also appeared more negative than 2D values (–20.8±5.4% vs. –19.2±5.0%), this difference did not reach statistical significance (P=0.051), with a similar correlation (r=0.48) (Table 3).

### Diagnostic Utility of 2D and 3D Echocardiographic Right Ventricular Strain in Identifying Subclinical RV Dysfunction in Repaired Tetralogy of Fallot

In patients with rTOF, all right ventricular (RV) strain parameters measured by both 2D and 3D echocardiography were significantly lower in the group with RV systolic dysfunction (defined as CMR RVEF<48%) compared to those with

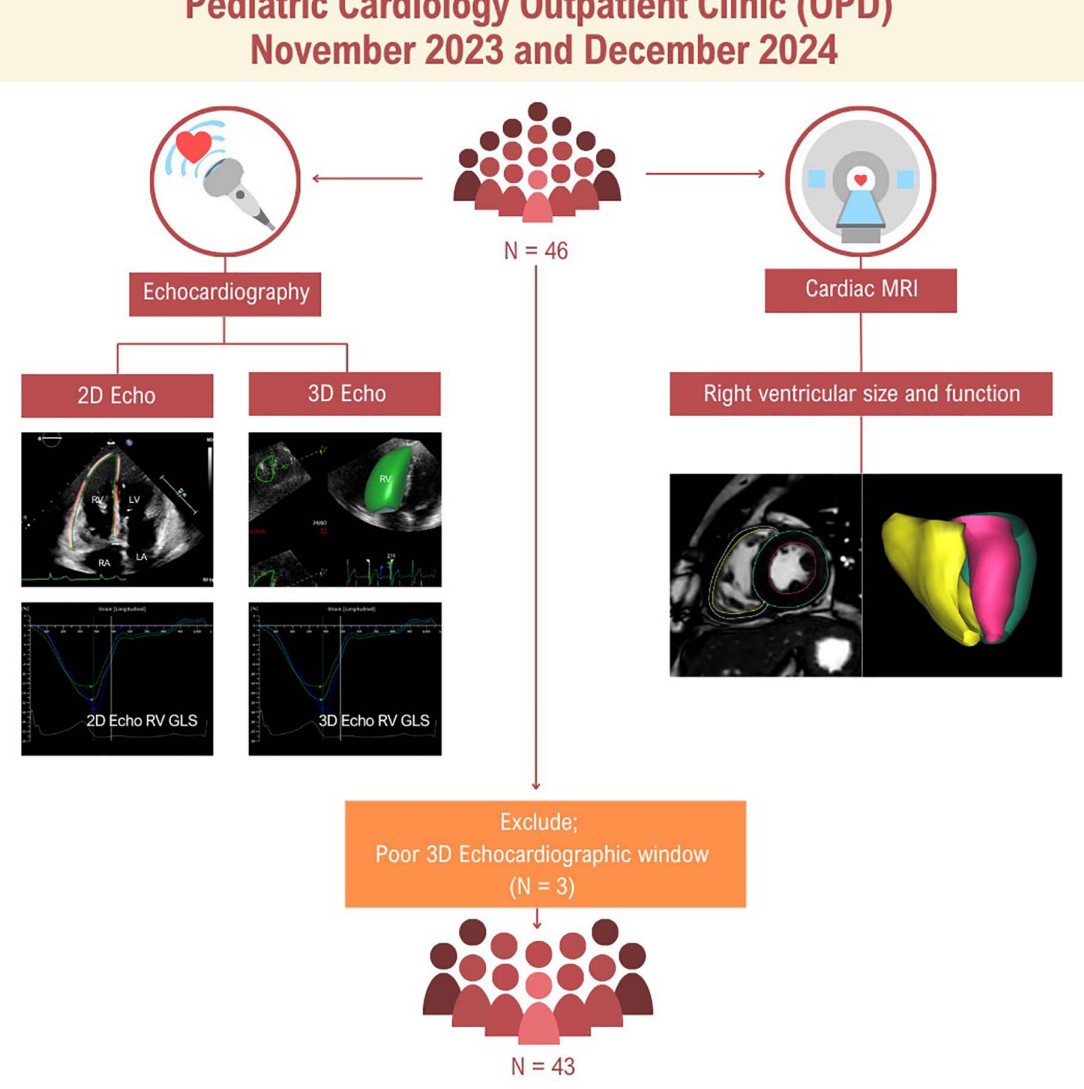

**Fig 1. Study Flow Diagram and Imaging Protocol.**

preserved RV function (RVEF ≥ 48%). The 2D RV-GLS was markedly reduced in the dysfunction group (−14.5 ± 4.1% vs. −19.5 ± 4.3%, P = 0.003), with an AUC of 0.79 (95% CI: 0.63–0.95). Similarly, the 2D RV free wall strain was significantly lower (−14.8 ± 4.4% vs. −20.3 ± 4.5%, P = 0.002), and 2D septal strain also showed a significant difference (−14.1 ± 4.7% vs. −18.7 ± 4.5%, P = 0.010), with AUCs of 0.78 and 0.75, respectively. For 3DE, the RV-GLS showed a significant difference between groups (−15.8 ± 3.6% vs. −21.5 ± 3.8%, P = 0.002), with an AUC of 0.86 (95% CI: 0.70–1.00). The 3D RV free wall strain provided the greatest discriminatory power with a mean difference of −7.4% (95% CI: −10.9 to −4.0, P < 0.001) and an AUC of 0.86 (95% CI: 0.72–1.00). Additionally, the 3D RV septal strain also differed significantly between groups (−16.7 ± 4.5% vs. −20.7 ± 4.8%, P = 0.029), with an AUC of 0.76 (95% CI: 0.56–0.96) (Table 4 and Fig 3).

Receiver operating characteristic (ROC) curves demonstrating the discriminative performance of echocardiographic parameters for identifying right ventricular systolic dysfunction (RVEF < 48%) in patients with rTOF. The area under ROC

**Table 1. Baseline characteristics of study population.**

| Characteristic | n = 43 (%) |
|---|---|
| Male | 27 (62.8%) |
| Age at study (years) | 17.9±5.8 |
| Height (cm) | 155.6±12.9 |
| Weight (kg) | 48.5±17.5 |
| BSA (per 1.73 m²) | 1.43±0.30 |
| BMI (kg/m²) | |
| < 18 | 20 (46.5%) |
| 18-25 | 18 (41.9%) |
| >25 | 5 (11.6%) |
| Diagnosis | |
| TOF with pulmonary stenosis | 38 (88.4%) |
| TOF with pulmonary atresia | 4 (9.3%) |
| TOF with absence of pulmonary artery | 1 (2.3%) |
| Palliative shunt, n | 14 (32.%) |
| Type of palliative shunt | |
| mBT shunt | 11 (78.5%) |
| Central shunt | 1 (7.1%) |
| Bilateral mBT shunt | 1 (7.1%) |
| mBT shunt and central shunt | 1 (7.1%) |
| Go for definite surgical repair | 29 (67.4%) |
| Age at first palliative shunt (month), median (IQR) | 15.5 (1, 20) |
| Type of total correction | |
| Transannular patch (TAP) | 34 (79.1%) |
| Non- transannular patch (valve sparing) | 6 (14.0%) |
| Right ventricle to pulmonary artery conduit | 3 (6.9%) |
| Age of total correction (month), median (IQR) | 40 (31, 49) |
| Syndromic | |
| None | 39 (90.7%) |
| 22q11 Deletion Syndrome | 2 (4.7) |
| VACTREL | 2 (4.7) |
| Heart failure symptoms | 0 |

Abbreviation: **BMI:** Body Mass Index; **BSA:** Body Surface Area; **IQR:** Interquartile Range; **mBT shunt:** Modified Blalock–Taussig shunt; **RV-PA conduit:** Right Ventricle to Pulmonary Artery Conduit; **TAP:** Transannular Patch; **TOF:** Tetralogy of Fallot

curve (AUC) for RVEF alone (A) was 0.85 (95% CI: 0.64–0.93). The addition of 2D right ventricular global longitudinal strain (RV-GLS) (B) improved discrimination with an AUC of 0.90 (95% CI: 0.57–0.97). Further inclusion of 3D RV-GLS (C) enhanced diagnostic accuracy with an AUC of 0.92 (95% CI: 0.84–0.99). Although, the combined model incorporating RVEF, 2D RV-GLS, and 3D RV-GLS achieved the highest diagnostic performance with an AUC of 0.93 (95% CI: 0.83–0.99) (Fig 4), The P value for the comparison between RVEF alone and the combined all parameters was not statistically significant.

## Discussion

This study demonstrates the clinical utility of 3DE for the assessment of RV size, function, and myocardial deformation in patients with rTOF. Our findings contribute to the existing body of evidence by comparing 2D and 3D echocardiographic

**Table 2. Correlation and agreement between 3D echocardiography and CMR parameters in patients with Repaired Tetralogy of Fallot.**

| n = 43 | 3D echo | CMR | P value | Correlation Coefficient | Mean bias (LOA) | P10 (95% CI) |
|---|---|---|---|---|---|---|
| RVEDV (mL) | 182.5±58.3 | 198.8±61.8 | **<0.001** | 0.95 | −16.3 (−55.4, 22.9) | 69.8 (53.9, 82.8) |
| RVEDVi (mL/m²) | 128.0±36.5 | 139.6±36.5 | **<0.001** | 0.91 | −11.3 (−40.8, 18.2) | 74.4 (58.8, 86.5) |
| RVESV (mL) | 99.4±37.9 | 100.1±37.8 | 0.765 | 0.90 | −0.8 (−33.2, 31.7) | 63.8 (46.7, 77.0) |
| RVESVi (mL/m²) | 68.1±22.3 | 70.1±23.1 | 0.218 | 0.897 | −2.0 (−22.2, 18.3) | 62.8 (46.7, 77.0) |
| RVEF (%) | 46.4±8.8 | 50.3±7.3 | **0.002** | 0.518 | −4.0 (−19.7, 12.5) | 51.2 (35.5, 66.7) |

**Abbreviation: 3D echo:** Three-Dimensional Echocardiography; **CMR:** Cardiovascular Magnetic Resonance Imaging; **CI:** Confidence Interval; **LOA:** Limits of Agreement; **P10:** Percentage of values within 10% of the reference standard; **RVEDV:** Right Ventricular End-Diastolic Volume; **RVEDVi:** Right Ventricular End-Diastolic Volume indexed to Body Surface Area; **RVEF:** Right Ventricular Ejection Fraction; **RVESV:** Right Ventricular End-Systolic Volume; **RVESVi:** Right Ventricular End-Systolic Volume indexed to Body Surface Area

parameters with CMR derived right ventricular function, supporting the complementary role of 3DE in the evaluation of right ventricular performance.

### 3D Echocardiography vs. CMR for Ventricular Volume and Function Assessment

In our cohort, 3DE demonstrated a strong association with CMR for right ventricular volumetric measurements. While correlation analysis showed a strong linear relationship between 3DE- and CMR-derived RV volumes, agreement between the two modalities was more appropriately assessed using Bland–Altman analysis. This analysis demonstrated a modest mean bias with acceptable limits of agreement, supporting the clinical feasibility of 3DE for RV volumetric assessment in routine practice [21–23]. However, a consistent trend of underestimating RV volumes using echocardiographic techniques was noted, which aligns with previous reports attributed to limitations in endocardial border delineation inherent in echocardiographic modalities CMR [21,24,25]. Despite these discrepancies, the magnitude of underestimation was small and remained within clinically acceptable limits for longitudinal follow-up in patients with rTOF. The RVEF derived from 3DE was significantly lower than that obtained from CMR [21]. In contrast with the prior study, RVEF was slightly overestimated in 3DE. This may a well-documented limitation of 3DE, influenced by its dependence on acoustic windows and potential foreshortening, emphasizing that CMR remains superior for precise RV functional quantification [4]. Nonetheless, the moderate correlation observed indicates the potential for 3DE to be useful for identifying patients with clinically relevant RV dysfunction, especially in settings where CMR availability is limited.

### Two-dimensional and 3D Right Ventricular Myocardial Strain for rTOF by Echocardiography Assessment

In our study, 3DE derived RV GLS demonstrated a moderate association with 2D RV GLS. The systematically more negative GLS values observed with 3DE reflect methodological differences between 2D and 3D speckle-tracking approaches, including the incorporation of out-of-plane myocardial motion in 3DE, rather than superior performance of one technique over the other. Therefore, RV GLS values should be interpreted in a modality-specific manner and should not be directly compared across imaging techniques [23,26–28].

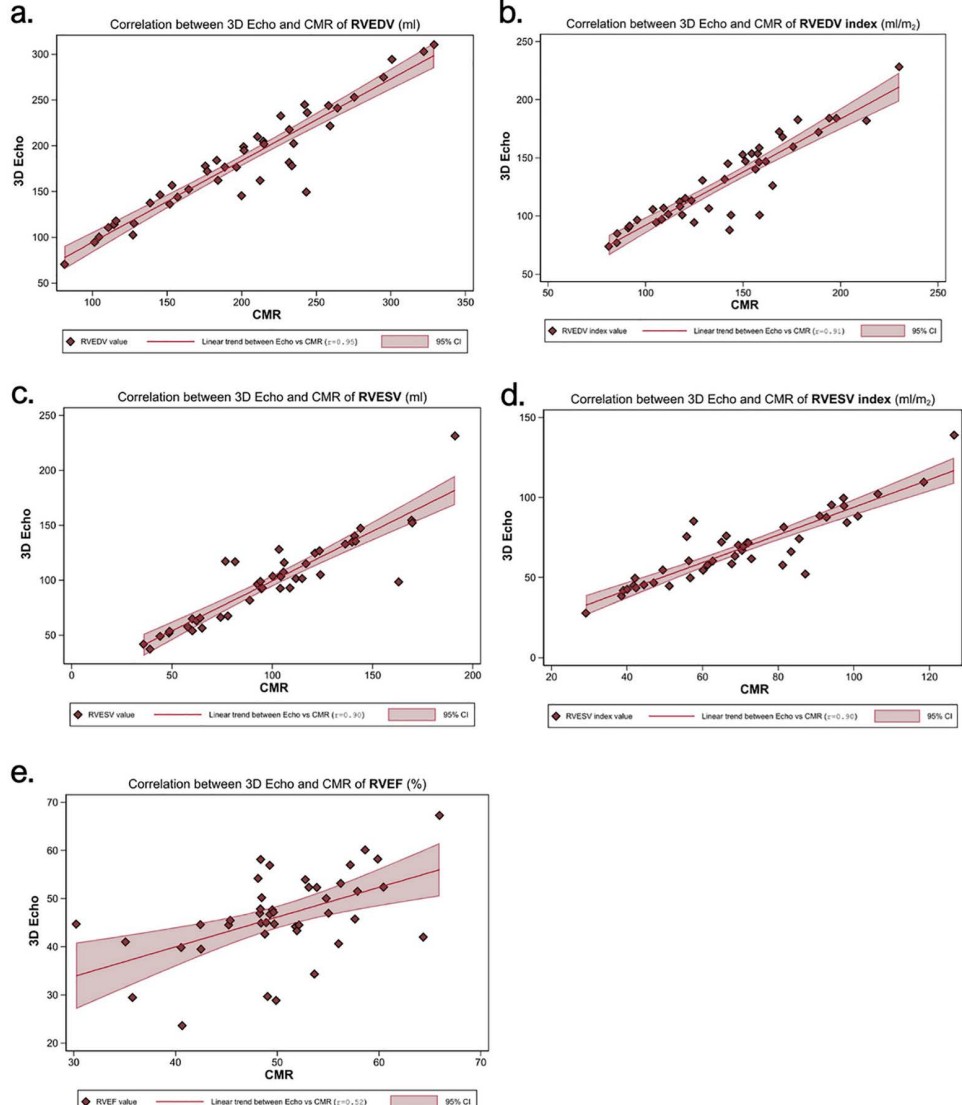

**Fig 2. Correlation Between Three-Dimensional Echocardiography (3DE) and Cardiovascular Magnetic Resonance (CMR) for Right Ventricular Volume and Function Parameters.** In each panel, individual patient values are shown as diamonds, the solid line denotes the linear regression fit with Pearson correlation coefficient **(r)**, and the shaded band represents the 95% confidence interval. **(a)**: Correlation of RV end-diastolic volume (RVEDV) between 3D echo and CMR (r = 0.95). **(b)**: Correlation of RVEDV indexed to body surface area (RVEDVi) between 3D echo and CMR (r = 0.91). **(c)**: Correlation of RV end-systolic volume (RVESV) between 3D echo and CMR (r = 0.90). **(d)**: Correlation of RVESV indexed to body surface area (RVESVi) between 3D echo and CMR (r = 0.90). **(e)**: Correlation of right ventricular ejection fraction (RVEF) between 3D echo and CMR (r = 0.52).

### Diagnostic Utility of 2D and 3D Echocardiographic Right Ventricular Strain in Identifying Subclinical RV Dysfunction in rTOF

When stratified by RVEF ≥48% vs. <48%, patients with reduced RVEF exhibited significantly less negative strain values across both 2D and 3D modalities, supporting the role of myocardial strain as an early marker of RV dysfunction [8,15,29]. A high AUC indicates the ability of a parameter or tool to discriminate patients who have or don't have RVEF <48%. Notably, 3DE-derived RV-GLS exhibited a superior trend of AUC for identifying this outcome. compared to 2DE. This suggests that 3DE-derived strain

**Table 3. Comparison and Correlation Between 2D and 3D Echocardiographic Assessment in Patients with Repaired Tetralogy of Fallot.**

| | Two dimensions (2D) | Three dimensions (3D) | P-value | Correlation coefficient |
|---|---|---|---|---|
| RV-GLS Echo | −18.5±4.7 | −20.3±4.4 | **0.009** | 0.52 |
| RV-LS free wall Echo | −19.2±5.0 | −20.8±5.4 | 0.051 | 0.48 |
| RV-LS septal Echo | −17.8±4.9 | −19.9±5.0 | **0.008** | 0.48 |

Abbreviations: **2D:** Two-Dimensional; **3D:** Three-Dimensional; **CMR:** Cardiovascular Magnetic Resonance Imaging; **Echo:** Echocardiography; **LV-GLS:** Left Ventricular Global Longitudinal Strain; **P-value:** Probability Value **RV-GLS:** Right Ventricular Global Longitudinal Strain; **RV-LS free wall:** Right Ventricular Longitudinal Strain of the Free Wall; **RV-LS septal:** Right Ventricular Longitudinal Strain of the Septal Wall

**Table 4. Comparison of Echocardiographic and CMR Parameter in TOF-Repaired Patients with and without RV Systolic Dysfunction.**

| | CMR RVEF <48% N=9 | CMR RVEF ≥48% N=34 | Mean Different (95% CI) | P value | AUC (95% CI) |
|---|---|---|---|---|---|
| RV-GLS echo 2D | −14.5±4.1 | −19.5±4.3 | −5.1 (−8.3, −1.8) | **0.003** | 0.79 (0.63, 0.95) |
| RV-LS free wall echo 2D | −14.8±4.4 | −20.3±4.5 | −5.6 (−8.9, −2.2) | **0.002** | 0.78 (0.60, 0.96) |
| RV-LS septal echo 2D | −14.1±4.7 | −18.7±4.5 | −4.6 (−8.0, −1.1) | **0.010** | 0.75 (0.57, 0.93) |
| RV-GLS echo 3D | −15.8±3.6 | −21.5±3.8 | −5.7 (−8.6, −2.9) | **0.002** | 0.86 (0.70, 1.00) |
| RV-LS free wall echo 3D | −14.9±5.0 | −22.4±4.4 | −7.4 (−10.9, −4.0) | **<0.001** | 0.86 (0.72, 1.00) |
| RV-LS septal echo 3D | −16.7 4.5 | −20.7 4.8 | −4.0 (−7.6, −0.4) | **0.029** | 0.76 (0.56, 0.96) |

Abbreviations: **2D:** Two-Dimensional; **3D:** Three-Dimensional; **95% CI**, 95% confident interval; **AUC**, area under receiver operating characteristic; **CMR:** Cardiovascular Magnetic Resonance Imaging; **Echo:** Echocardiography; **LV-GLS:** Left Ventricular Global Longitudinal Strain; **P-value:** Probability Valueꓵ **RV-GLS:** Right Ventricular Global Longitudinal Strain; **RV-LS free wall:** Right Ventricular Longitudinal Strain of the Free Wall; **RV-LS septal:** Right Ventricular Longitudinal Strain of the Septal Wall.

analysis potentially serves as a non-invasive tool for early identification of ventricular dysfunction in rTOF patients, potentially reducing the need for frequent CMR evaluations [13]. Interestingly, in patients with rTOF, RV GLS is reduced even when the RVEF measured by CMR remains within the normal range. This suggests that RV GLS may serve as a more sensitive marker for the early detection of subclinical right ventricular systolic dysfunction and may be used for predicting adverse outcomes, offering valuable insights beyond conventional volumetric assessment [30]. In our study, the highest diagnostic performance was observed when combining RVEF with both 2D and 3D RV-GLS, achieving an AUC of 0.93 (95% CI: 0.83–0.99). These findings underscore the superior discriminative ability of 3D RV-GLS over conventional volumetric indices and 2D strain alone. This is likely attributed to the ability of 3DE to capture the complex RV geometry more accurately, thereby allowing for a comprehensive analysis of myocardial deformation. Although a clear trend was observed, indicating the added diagnostic value of 3D RV-GLS, there is no statistical significance. Given its non-invasiveness and increasing accessibility, 3D RV-GLS may serve as a robust adjunctive tool in the longitudinal surveillance of TOF patients, particularly for the early detection of subclinical RV dysfunction and guiding timely interventions, such as pulmonary valve replacement [11].

## Clinical Implications

Three-dimensional echocardiography offers essential diagnostic and prognostic insights into RV function in patients with rTOF. Notably, our study demonstrates a strong agreement between 3DE and CMR for RV volumetric measurements,

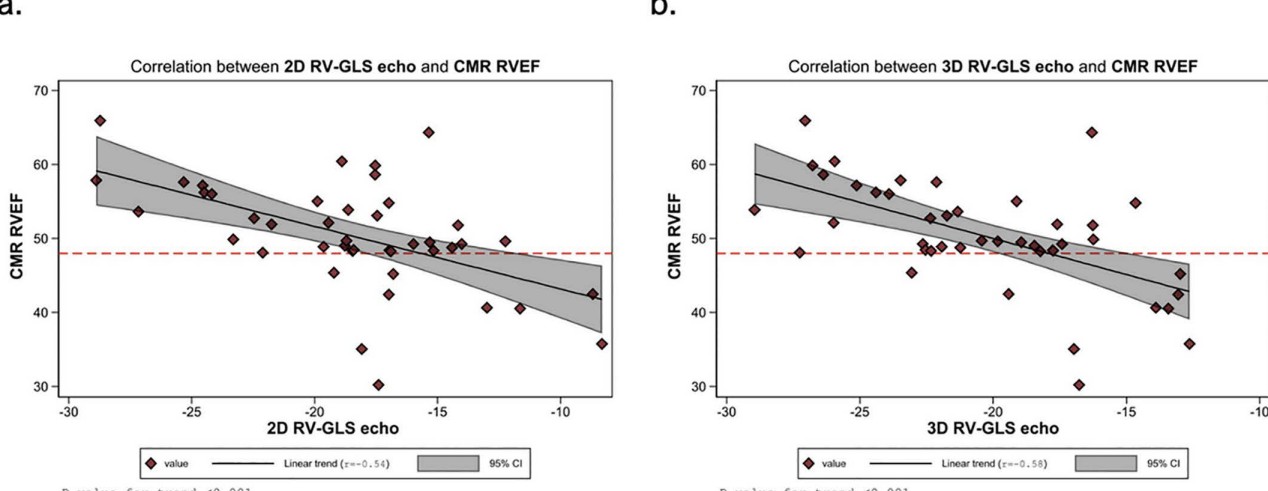

**Fig 3. Correlation between RV Global Longitudinal Strain (RV-GLS) and CMR-derived Right Ventricular Ejection Fraction (RVEF).** In each panel, individual patient values are shown as diamonds, the solid line denotes the linear regression fit with Pearson correlation coefficient **(r)**, and the shaded band represents the 95% confidence interval. The horizontal dashed line marks an RVEF of 48%. **(a)**: 2D echocardiography-derived RV-GLS versus CMR RVEF (r=−0.54). **(b)**: 3D echocardiography-derived RV-GLS versus CMR RVEF (r=−0.58).

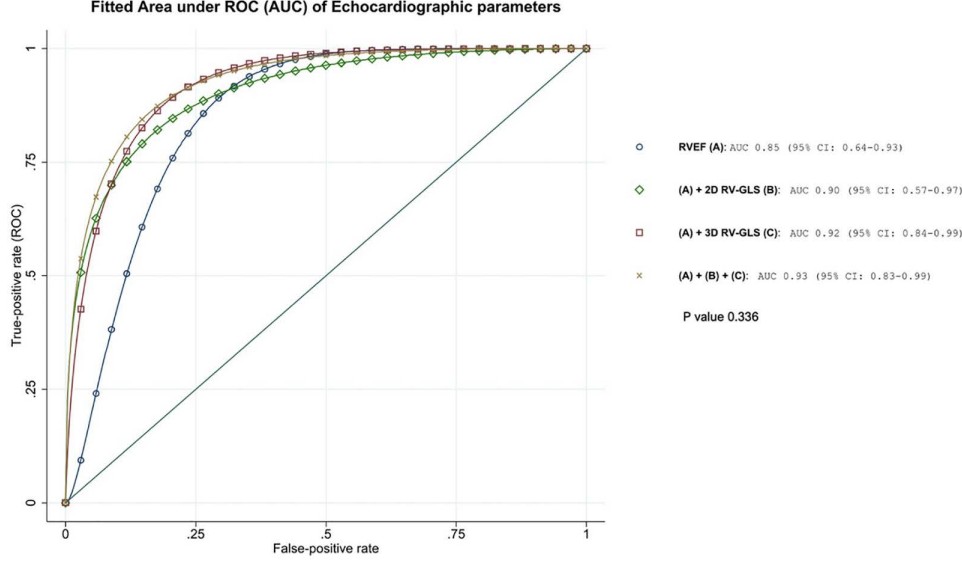

**Fig 4. Fitted Area under ROC curves (AUCs) illustrate the incremental discriminative performance of different echocardiographic (Echo) strain parameters when added to Echo RVEF for identifying right ventricular systolic dysfunction (RVEF < 48%) in patients with repaired Tetralogy of Fallot.**

reinforcing the reliability of 3DE a practical tool in clinical practice. Beyond volumetric assessment, 3DE provides the added advantage of myocardial deformation analysis, specifically RV GLS, which enhances the functional evaluation of RV performance. The ability of 3DE-derived strain to detect early signs of RV dysfunction, potentially before observable changes in ejection fraction, underscores its value in the longitudinal surveillance of rTOF patients.

Although 3DE is generally more accessible and cost-effective than CMR, its broader clinical implementation depends on equipment availability, validated analysis software, and operator expertise. Accurate acquisition and post-processing require trained personnel and vendor-specific platforms. In this study, right ventricular volumetric and strain analyses were performed using TomTec Imaging Systems, which has been widely validated for assessment of right ventricular size, function, and myocardial deformation in both congenital and non-congenital cardiac populations; however, measurements may not be directly interchangeable across different software platforms. Consistent with prior reports, 3DE demonstrated a tendency to underestimate right ventricular volumes compared with CMR. In our study, this underestimation was systematic rather than random, as indicated by a consistent negative mean bias on Bland–Altman analysis, likely related to challenges in endocardial border delineation and incomplete right ventricular coverage inherent to echocardiographic imaging.

Importantly, our study uniquely integrates simultaneous 2D, 3D echocardiographic-derived RV GLS and CMR assessments within the same cohort and imaging session. This comprehensive approach illustrates the feasibility of using 3DE as a robust and accessible modality for both serial volumetric and strain-based monitoring. The major clinical advantage of 3DE lies not in the small numerical difference in strain values, but in its ability to provide a comprehensive, integrated assessment of right ventricular size, systolic function, and myocardial deformation within a single imaging modality. This integrative approach strengthens the clinical relevance of 3DE beyond volume estimation and supports its potential role in guiding earlier intervention and follow-up strategies.

## Study Limitations

Several limitations of this study should be acknowledged. First, our sample size may be relatively small for some secondary objectives of this study. In addition, multiple hypothesis testing was performed, which increases the possibility that some statistically significant findings may have occurred by chance. The focus study to confirm those findings in our study is warranted. Second, image quality remains a challenge for 3DE, particularly in patients with suboptimal acoustic windows or marked right ventricular dilation. In such cases, incomplete right ventricular coverage within a single acquisition may lead to systematic underestimation of volumetric and functional parameters, reducing measurement reliability. Consequently, patients with very advanced right ventricular remodeling, significant residual shunts, or arrhythmias were excluded, and our findings may not be directly applicable to these clinically important subgroups, for whom CMR imaging remains the reference standard for accurate assessment of right ventricular size and systolic function. Third, echocardiographic strain measurements may be influenced by vendor-specific analysis algorithms, which could affect reproducibility across platforms. In addition, the use of a single experienced assessor may raise concerns regarding potential observer bias, although standardized post-processing protocols and blinding between echocardiographic and CMR analyses were applied to mitigate this risk. Furthermore, only moderate correlations were observed between 2D and 3D strain parameters, highlighting modality-specific differences and underscoring the importance of interpreting strain values within the context of the imaging technique used. Despite these limitations, the study reflects real-world clinical practice and benefits from standardized same-day echocardiography and CMR acquisition, minimizing physiologic variability. Future larger, multicenter studies with multiple independent readers are warranted to validate the reproducibility of 3DE-derived parameters and to define further their role in risk stratification and long-term outcome assessment in patients with rTOF.

## Conclusion

Three-dimensional echocardiography demonstrates a consistent association with CMR for the assessment of right ventricular volumes but modest underestimation of volumetric and functional parameters. Abnormal right ventricular strain was observed in patients with CMR-defined systolic dysfunction, supporting the clinical relevance of strain analysis. Overall, 3DE may serve as a feasible complementary tool for longitudinal right ventricular assessment in patients with rTOF, alongside CMR as the reference standard.

**Disclosure of Artificial Intelligence (AI) Programs:** This work did not use any artificial intelligence program for data analysis or manuscript writing.

## Supporting information

**S1 Fig. Bland–Altman Analysis Comparing three-dimensional echocardiography (3D Echo) and cardiovascular magnetic resonance (CMR) Measurements of Right Ventricular (RV) Size and Function.** In each panel, the y-axis shows the bias (3D Echo – CMR) plotted against the corresponding CMR measurement on the x-axis. The solid blue line is the mean bias; the pink shaded band represents its 95% confidence interval (CI). Red dashed lines indicate the limits of agreement (LOA), and red dotted lines indicate the 95% CI around each LOA. **(a)**: RV end-diastolic volume (RVEDV). **(b)**: RVEDV indexed to body surface area (RVEDVi). **(c)**: RV end-systolic volume (RVESV). **(d)**: RVESV indexed to body surface area (RVESVi). **(e)**: RV ejection fraction (RVEF).
(PNG) (Add Supplymentary Fig)

## Author contributions

**Conceptualization:** Kwannapas Saengsin.

**Data curation:** Kwannapas Saengsin.

**Formal analysis:** Pakpoom Wongyikul, Phichayut Phinyo.

**Investigation:** Kwannapas Saengsin, Kamonchanok Intamul, Jinnawat Rattanang, Patanin Chindarungrueangkun.

**Methodology:** Kwannapas Saengsin, Pakpoom Wongyikul, Phichayut Phinyo.

**Supervision:** Kwannapas Saengsin.

**Writing – original draft:** Kwannapas Saengsin, Pakpoom Wongyikul.

**Writing – review & editing:** Kwannapas Saengsin, Rekwan Sittiwangkul, Seehapong Petcharat, Ashwin Prakash.

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
