## [Decision Letter · Decision Letter 0]

12 Jan 2026

Dear Dr. Saengsin,

Thank you for submitting your manuscript to PLOS ONE. After careful consideration, we feel that it has merit but does not fully meet PLOS ONE’s publication criteria as it currently stands. Therefore, we invite you to submit a revised version of the manuscript that addresses the points raised during the review process.

We look forward to receiving your revised manuscript.

Kind regards,

Atnafu Mekonnen Tekleab, M.D, MPH

Academic Editor

PLOS One

Journal Requirements:

5. We note that Figure 1 in your submission contain copyrighted images. All PLOS content is published under the Creative Commons Attribution License (CC BY 4.0), which means that the manuscript, images, and Supporting Information files will be freely available online, and any third party is permitted to access, download, copy, distribute, and use these materials in any way, even commercially, with proper attribution. For more information, see our copyright guidelines: http://journals.plos.org/plosone/s/licenses-and-copyright.

Additional Editor Comments:

I would like to thank the authors for their very interesting and clinically relevant research work. In addition to addressing the comments raised by the reviewers, I would like to request the authors to address the following issues. Particularly, I would like to invite the authors special attention related to the methods and conclusion section of the study since there are major issues raised by the reviewers.

Line 145-146: “The cardiac function was evaluated by a single experienced imaging pediatric cardiologist (KS).” I assume that KS is the first author of this manuscript. If that is the case, isn’t there an observer bias? How about the impact of using single assessor on the impact of the accuracy of the data set?Lines 168- 172: what was the basis to determine sample size by taking the mean bias of 5 and SD of 8? Do you think these values help to produce a stable model? Needs elaboration!Line 168: sample size calculated using pilot study. Why sample was not calculated based on previous study findings?Lines 171- 172: States that “calculations were performed using a two sampled-paired t-test”.  However, comparison of RVEF by 3DE vs CMR is agreement comparison, not a difference in means comparison. Please explain this.Lines 162- 163: What is the importance of using LOA alongside P10? Do the two parameters convey different info?Lines 186-192:Mean bias is reported, but 95% limits of agreement are missing.RVESVi correlation is stated as interval (r = 0.90–0.88) and its bias is described as “minimal”. What odes “minimal bias” means?Lines 196- 204: Please, explain the clinical relevance of such a small difference between the 2D and 3D GLS of RV strain values though the two have a statistically significant difference.Lines 232- 235: it appears an overstated claim.Lines 237- 252: The discussion mentions the “strong correlation” between 3DE and MRI as an indicator of agreement between the two diagnostic modalities. However, correlation indicates association not agreement between two diagnostic modalities. Please check and revise accordingly as it may mislead readers.Lines 256- 259: The authors stated that “*Our study demonstrated that 3DE RV GLS has a moderate correlation with 2DE GLS. Three-DE RV GLS had significantly more negative values than 2DE-derived RV-GLS. This suggests that 3D RV GLS imaging may provide a more sensitive detection of RV myocardial deformation compared to 2D assessment..”.*  It may be misleading to infer that the 3DE RV GLS has more negative values than 2DE RV-GLS and that implies better performance of 3DE RV GLS. Please, check it and revise accordingly.Lines 277- 283: The authors attributed the lack of statistical significance to the “small sample size”. However, you also claimed that the study was adequately powered (power 90%). Hence, the two ideas seem to contradict.Lines 305- 316: The limitations of the study mentioned under this paragraph appear to be general rather than specific to their study. Here are the reasons: small sample size is mentioned as limitation while you already told us in the methods section that the study was adequately powered; image quality is mentioned as limitation for 3DE while you told us that you have excluded cases that had poor image quality; result differences based on vendor specific strain analysis is mentioned as limitation though that was not related to the current study. Please clarify or revise accordingly.Does good AUC imply good agreement between 3DE and CMR and vise-versa? Also, the CMR is both used as gold standard and comparator. Give us short explanation in the discussion.

Reviewers' comments:

Reviewer's Responses to Questions

**Comments to the Author**

1. Is the manuscript technically sound, and do the data support the conclusions?

Reviewer #1: Partly

Reviewer #2: Yes

2. Has the statistical analysis been performed appropriately and rigorously?

Reviewer #1: Yes

Reviewer #2: Yes

3. Have the authors made all data underlying the findings in their manuscript fully available?

Reviewer #1: Yes

Reviewer #2: No

4. Is the manuscript presented in an intelligible fashion and written in standard English?

Reviewer #1: Yes

Reviewer #2: Yes

Reviewer #1: Thank you for the opportunity to review the paper.

1. My general comments about the research work

The research tries to address a clinically relevant and challenging long-term issue in patients who had undergone surgical repair for Tetralogy of Fallot (both staged and primary repair).The study design is appropriate with detailed imaging protocols and a gold standard (CMR). The attempt to minimize physiologic variability impacts by doing the imaging studies on the same day is another strength of the study. However, in view of the modest correlation observed between 3D ECHO and CMR in quantification of right ventricular systolic function, the clinical implication of the research findings is slightly overstated.

2. Specific comments for improvement

• Methods section: whether the 2D/3D echocardiographer is blinded from the CMR findings and vise versa is unclear to avoid bias.

• Discussion: The study fails to sufficiently explain/discuss the discrepancy between the 3D ECHO and CMR with respect to RV systolic function assessment. Due to the modest correlation between 3D ECHO and CMR in RV systolic function assessment in the study, the first statement on clinical implications appear to be slightly overstated.

Reviewer #2: This study addresses a significant clinical issue in adult congenital heart disease (ACHD), specifically the onset of right ventricular (RV) systolic dysfunction following the repair of Tetralogy of Fallot. It introduces an alternative method for assessing RV volume and function in these patients, aside from the gold standard of cardiac magnetic resonance imaging (CMR). The research also compares the commonly used two-dimensional echocardiography (2DE) for evaluating RV volumes and myocardial deformation in patients with repaired Tetralogy of Fallot. The study met its power calculation requirements (29 required, 46 enrolled, 43 analyzed), which enhances the credibility of its findings. By defining RV dysfunction as an RV ejection fraction (RVEF) of less than 48%, the study establishes a clear threshold for diagnostic utility analysis. The use of Bland-Altman plots to evaluate bias and receiver operating characteristic (ROC) curves (area under the curve, AUC) to assess diagnostic accuracy is an appropriate statistical approach for a validation study. The writing maintains a professional tone and adheres to standard medical reporting. The results logically progress from baseline characteristics to direct comparisons of volumes, strain parameters, and ultimately to diagnostic performance.

The authors indicate that the study data cannot be publicly shared due to institutional regulations. However, data can be accessed through the Institutional Ethics Committee at the Faculty of Medicine, Chiang Mai University, for researchers who meet confidentiality criteria. This approach limits the availability of the study data.

The research was conducted between November 2023 and December 2024, but ethical approval was obtained from the Institutional Review Board at the Faculty of Medicine of Chiang Mai University in June 2025. Typically, ethical approval should precede data collection.

Patients with suboptimal 3D echocardiographic images (such as those with large RVs) as well as those with significant residual shunts and arrhythmias were excluded from the study. This exclusion is noteworthy, as these patients are crucial for integrating imaging modalities into clinical care and assessing the need for follow-up procedures like pulmonary valve replacements.

While the study's title indicates that it includes patients with repaired Tetralogy of Fallot, it is important to note that 32.6% of participants were only palliated. This subgroup may exhibit different pressure and volume dynamics affecting their right ventricles compared to fully repaired patients, given that surgical repair influences RV volume, function, and strain patterns. Furthermore, a correlation could have been better established between the timing of the study and the duration since repair.

Although CMR is generally more expensive and technically complex, three-dimensional echocardiography (3DE) may be more accessible and cost-effective. However, the study should address considerations regarding cost, required technical expertise, and the availability of 3DE software for broader implementation. Additionally, it would be beneficial to discuss the validity and applicability of TomTec Imaging Systems, which is used for this study, for assessing RV volume, function, and myocardial strains in non-cardiac patients.

Finally, the authors acknowledge that 3DE tends to underestimate volumes compared to CMR. It would be useful to include a brief commentary on whether this underestimation is systematic (consistently lower) or random, as systemic bias could be corrected in clinical practice using an adjustment factor.

**Do you want your identity to be public for this peer review?** For information about this choice, including consent withdrawal, please see our Privacy Policy

Reviewer #1: No

Reviewer #2: No

---

## [Author Response · Author response to Decision Letter 1]

13 Feb 2026

Utility of Three-Dimensional Echocardiography for Evaluating Right Ventricular Size and Function and Ventricular Myocardial Deformation in Repaired Tetralogy of Fallot

Thank you very much for your kind considering on our manuscript. We appreciate the constructive and very helpful feedback and the supportive comments from the reviewers. We have carefully considered the reviewers’ comments and suggestions and have edited the manuscript to address them. We believe the manuscript has been greatly improved after the revision based on those suggestions.

We have addressed each comment as outlined below with the yellow color highlight in the manuscript

1.Please ensure that your manuscript meets PLOS ONE's style requirements, including those for file naming.

Answer: Our manuscript meets PLOS ONE's style requirements

Answer: Remove funding-related text from the manuscript.

Answer: The data underly¬ing this study cannot be made publicly available due to patient confidentiality restrictions. However, the data will be made available to qualified researchers upon reasonable request. Data requests may be directed to the Research Ethics Committee of the Faculty of Medicine, Chiang Mai University which is responsible for oversight of data governance and access. Please contact: Research Ethics Committee, Faculty of Medicine, Chiang Mai University Address: Rianroum Buiding, 3rd floor Faculty of Medicine, Chiang Mai University 110 Intavaroros Road, Tambon Sriphum, Amphoe Muang, Chiang Mai Thailand 50200 Business Tel.: No. +6653936643, +6653936641, +6653935279 ext. 22-23 Email: research.med.cmu@gmail.com, researchmed@cmu.ac.th. This committee is independent of the study authors and is authorized to review and approve data access requests in accordance with insti¬tutional policies and ethical guidelines.

Answer: Ethics statement only appears in the Methods section of your manuscript.

5. We note that Figure 1 in your submission contain copyrighted images. All PLOS content is published under the Creative Commons Attribution License (CC BY 4.0), which means that the manuscript, images, and Supporting Information files will be freely available online, and any third party is permitted to access, download, copy, distribute, and use these materials in any way, even commercially, with proper attribution. For more information, see our copyright guidelines: http://journals.plos.org/plosone/s/licenses-and-copyright.

Answer: Thank you very much for your careful review and for raising this important copyright consideration. We would like to clarify that all echocardiography and cardiac MRI images presented in Figure 1 are original images generated by the authors from our own patient data, acquired and analyzed at our institution as part of routine clinical care and this research study. These images were not reproduced, adapted, or reprinted from any previously published sources, textbooks, software manuals, or proprietary promotional materials.Accordingly, the authors hold the copyright to these images and confirm that no third-party copyrighted material is included in Figure 1. The figure is therefore eligible to be published under the Creative Commons Attribution License (CC BY 4.0), in full compliance with PLOS ONE’s open-access policy, and no additional permission is required. To further ensure clarity and transparency, we have updated the figure caption to explicitly state that Figure 1 is an author-created figure using original imaging data. We sincerely appreciate the editorial team’s guidance and the opportunity to clarify this matter.

Answer: We appreciate the reviewer’s thoughtful suggestions, which helped us reassess and strengthen the contextual foundation of our work.

Additional Editor Comments:

I would like to thank the authors for their very interesting and clinically relevant research work. In addition to addressing the comments raised by the reviewers, I would like to request the authors to address the following issues. Particularly, I would like to invite the authors special attention related to the methods and conclusion section of the study since there are major issues raised by the reviewers.

• Line 145-146: “The cardiac function was evaluated by a single experienced imaging pediatric cardiologist (KS).” I assume that KS is the first author of this manuscript. If that is the case, isn’t there an observer bias? How about the impact of using single assessor on the impact of the accuracy of the data set?

Answer: Thank you for this important and thoughtful comment. Yes, KS is the first author of this manuscript and served as the primary imaging cardiologist for cardiac magnetic resonance (CMR) analysis. We acknowledge that the use of a single assessor may raise concerns regarding potential observer bias. However, this approach reflects the real-world clinical practice and resource limitations in Northern Thailand, where there is currently only one pediatric cardiologist with specialized expertise in congenital cardiac MRI interpretation. In this setting, the use of a single assessor is unavoidable and represents standard clinical practice. Importantly, KS holds Society for Cardiovascular Magnetic Resonance (SCMR) Level III certification, which represents the highest level of formal training and competency in CMR acquisition, interpretation, and reporting. This level of expertise helps ensure high-quality and reliable measurements. To further minimize potential observer bias, all CMR analyses were performed using standardized, validated post-processing protocols, and the assessor was blinded to echocardiographic strain results and relevant clinical data at the time of analysis. In addition, measurements were obtained using semi-automated software with predefined contouring algorithms, reducing subjective influence on volumetric assessment. We acknowledge that the lack of inter-observer variability analysis is a limitation of the study, and this has now been explicitly addressed in the Study Limitations section.

“Several limitations of this study should be acknowledged. First, the relatively modest sample size may limit the generalizability of our findings and precluded robust subgroup analyses, including stratification by surgical strategy or duration since definitive repair. Second, image quality remains a challenge for 3DE, particularly in patients with suboptimal acoustic windows or marked right ventricular dilation. In such cases, incomplete right ventricular coverage within a single acquisition may lead to systematic underestimation of volumetric and functional parameters, reducing measurement reliability. Consequently, patients with very advanced right ventricular remodeling, significant residual shunts, or arrhythmias were excluded, and our findings may not be directly applicable to these clinically important subgroups, for whom CMR imaging remains the reference standard for accurate assessment of right ventricular size and systolic function. Third, echocardiographic strain measurements may be influenced by vendor-specific analysis algorithms, which could affect reproducibility across platforms. In addition, the use of a single experienced assessor may raise concerns regarding potential observer bias, although standardized post-processing protocols and blinding between echocardiographic and CMR analyses were applied to mitigate this risk. Furthermore, only moderate correlations were observed between 2D and 3D strain parameters, highlighting modality-specific differences and underscoring the importance of interpreting strain values within the context of the imaging technique used. Despite these limitations, the study reflects real-world clinical practice and benefits from standardized same-day echocardiography and CMR acquisition, minimizing physiologic variability. Future larger, multicenter studies with multiple independent readers are warranted to validate the reproducibility of 3DE-derived parameters and to further define their role in risk stratification and long-term outcome assessment in patients with rTOF”.

Study limitation

Pages 18 and 19; lines 349-371

Page: 16, lines 312-314

• Lines 168- 172: what was the basis to determine sample size by taking the mean bias of 5 and SD of 8? Do you think these values help to produce a stable model? Needs elaboration!

Answer: We thank the reviewer for raising this important point. The justification for the assumed effect size of the mean bias and its variance was based on findings from our pilot study [1]. In addition, we accounted for within-subject correlation by assuming an intra-subject correlation coefficient of 0.5.

As our primary objective was to compare the mean right ventricular ejection fraction (RVEF) between echocardiography and cardiac magnetic resonance (CMR), we estimated the mean bias as the difference in mean RVEF obtained from the two modalities. Our intention was not to derive a precise or stable estimate, but rather to explore and describe patterns of disagreement between the measurement methods. Accordingly, residual and systematic differences were evaluated using the Bland Altman approach. Because our analysis was not based on a formal statistical parametric model, the sample size calculation derived from this approach should not lead to the concern raised by the reviewer

Reference

1. Shieh, G. (2013). On Using a Pilot Sample Variance for Sample Size Determination in the Detection of Differences between Two Means: Power Consideration. Psicologica, 34, 125-143.

• Line 168: sample size calculated using pilot study. Why sample was not calculated based on previous study findings?

Answer: We appreciate the reviewer’s comment. Sample size estimation is typically used to determine the acceptable statistical power for hypothesis testing based on an expected effect size. In the present study, we chose to base the effect size on our own pilot data, as we considered our study population to best reflect the variance and effect size in our target population [1], rather than relying on assumptions from other populations with potentially different characteristics.In addition, there is limited published evidence describing the mean bias of RVEF between echocardiography and CMR in patients with tetralogy of Fallot (TOF).

Reference

1. Shieh, G. (2013). On Using a Pilot Sample Variance for Sample Size Determination in the Detection of Differences between Two Means: Power Consideration. Psicologica, 34, 125-143.

• Lines 171- 172: States that “calculations were performed using a two sampled-paired t-test”. However, comparison of RVEF by 3DE vs CMR is agreement comparison, not a difference in means comparison. Please explain this.

Answer: We appreciate the reviewer’s feedback and agree that mean bias alone does not fully capture agreement, as it reflects only the systematic difference between the two measurement methods [1]. Agreement between the modalities was therefore assessed using the Bland Altman method, which evaluates both systematic and residual differences. The calculation of the mean bias and its 95% confidence interval forms an integral part of the Bland Altman analysis [2]. We additionally included hypothesis testing to inform whether the observed systematic bias was statistically significant.

References

1. Bland, J. M., and D. G. Altman. 1986. Statistical methods for assessing agreement between two methods of clinical measurement. Lancet 327: 307–310. https://doi.org/10.1016/S0140-6736(86)90837-8.

2. Mokkink, L.B., Boers, M., van der Vleuten, C.P.M. et al. COSMIN Risk of Bias tool to assess the quality of studies on reliability or measurement error of outcome measurement instruments: a Delphi study. BMC Med Res Methodol 20, 293 (2020). https://doi.org/10.1186/s12874-020-01179-5

• Lines 162- 163: What is the importance of using LOA alongside P10? Do the two parameters convey different info?

Answer: The limits of agreement (LOA) inform us about the width of the 95% error range observed in the da

---

## [Editor Report · Decision Letter 1]

19 Feb 2026

Utility of Three-Dimensional Echocardiography for Evaluating Right Ventricular Size and Function and Ventricular Myocardial Deformation in Repaired Tetralogy of Fallot

PONE-D-25-55797R1

Dear Dr. Saegsin,

We’re pleased to inform you that your manuscript has been judged scientifically suitable for publication and will be formally accepted for publication once it meets all outstanding technical requirements.

Kind regards,

Atnafu Mekonnen Tekleab, M.D, MPH

Academic Editor

PLOS One
---

## [Editor Report · Acceptance letter]

PONE-D-25-55797R1

PLOS One

Dear Dr. Saengsin,

I'm pleased to inform you that your manuscript has been deemed suitable for publication in PLOS One. Congratulations! Your manuscript is now being handed over to our production team.

Kind regards,

on behalf of

Dr. Atnafu Mekonnen Tekleab

Academic Editor

PLOS One